# Synthesizing Adversarial Visual Scenarios for Model-Based Robotic Control

**Shubhankar Agarwal**⋆ **and Sandeep P. Chinchali** ⋆

**Abstract:** Today's robots usually interface data-driven perception and planning models with classical model-predictive controllers (MPC). Often, such learned perception/planning models produce erroneous waypoint predictions on out-of-distribution (OoD) or even adversarial visual inputs, which increase control cost. However, today's methods to train robust perception models are largely task-agnostic – they augment a dataset using random image transformations or adversarial examples targeted at the vision model in *isolation*. As such, they may introduce pixel perturbations that are ultimately benign for control. In contrast to prior work that synthesizes adversarial examples for single-step vision tasks, our key contribution is to synthesize adversarial scenarios tailored to multi-step, model-based control. To do so, we use differentiable MPC methods to calculate the sensitivity of a model-based controller to errors in state estimation. We show that *re-training* vision models on these adversarial datasets improves control performance on OoD test scenarios by up to 36.2% compared to standard task-agnostic data augmentation. We demonstrate our method on examples of robotic navigation, manipulation in RoboSuite, and control of an autonomous air vehicle.

## 1  Introduction

Imagine a drone that must safely navigate using its camera and a learned perception module, such as a deep neural network (DNN). The drone's perception DNN and planning module output a sequence of waypoints that are tracked by MPC to achieve a low-cost trajectory [1, 2]. However, the drone will often observe weather and terrain conditions that are far from its original image training distribution. In this paper, we ask whether we can automatically synthesize adversarial visual scenarios for multi-step, *model-based* control tasks in order to re-train more robust robotic perception models.

Today's robust training methods include data augmentation [3, 4], domain randomization [5, 6], and adversarial training [7–9]. These methods add synthetic visual examples by randomly cropping, rotating, or adversarially altering targeted pixels in a training distribution. However, such augmentation methods aim to improve the vision model in isolation and hence are largely *task-agnostic*. While some methods [10–12] perform data augmentation for the ultimate end-to-end control task, the data-augmentation techniques are still task-agnostic, such as using random crops, blur, etc. As such, they often synthesize visual examples that do not adequately affect control-relevant states.

Our work bridges advances in generative models [13, 14], differentiable rendering [15, 16], and differentiable MPC [17]. These advances enable us to synthesize realistic visual scenes from a set of latent, often-interpretable, parameters and calculate their impact on control cost. Our principal contribution is a training procedure (Fig. 1) that efficiently perturbs the latent representation of an image so that we render scenarios that are poor for control, but consistent with naturally-occurring data. Moreover, we can visualize the latent representations of scenes to see how task-driven data augmentation improves the diversity of training examples, resulting in more robust models.

*Literature Review:*  Many perception models are susceptible to adversarial examples [7–9, 18]. For example, the Fast Gradient Sign Method (FGSM) calculates the sensitivity of a pre-trained classifier to distortions in the image input, which guides how we synthesize human-imperceptible pixel changes that cause mis-classifications. However, to the best of our knowledge, there is no work describing how such image perturbations ultimately affect *multi-step, model-based* control.

---

⋆Department of Electrical and Computer Engineering (ECE), The University of Texas at Austin, Austin, TX
`{somi.agarwal,sandeepc}@utexas.edu`

6th Conference on Robot Learning (CoRL 2022), Auckland, New Zealand.

Our work is distinct from robust adversarial reinforcement learning (RARL) [19–22]. First, these methods aim to improve a *control* policy, either by perturbing an agent's dynamics or introducing an adversarial RL agent that physically impedes progress. In contrast, we focus on fixed, model-based controllers (often represented by a convex program), and instead focus on re-training perception models whose state estimates serve as input parameters for convex MPC. To do so, we use differentiable MPC methods [17, 23, 24] to compute the sensitivity of the control cost to erroneous perceptual inputs. As such, we leverage the structure of MPC to efficiently synthesize adversarial inputs as opposed to model-free RARL methods.

As noted earlier, today's visual data augmentation [4] is largely task-agnostic – random image crops or rotations might alter perception waypoints, but barely affect control cost. Instead, our method leverages the MPC task's structure to targetedly perturb pixels that are most salient for control. Finally, our work differs from gradient-free Bayesian Optimization [25] methods (e.g., for safety assurances for self-driving cars [26, 27]) since we explicitly leverage the gradient of MPC's task cost. In light of this prior work, our contributions are as follows:

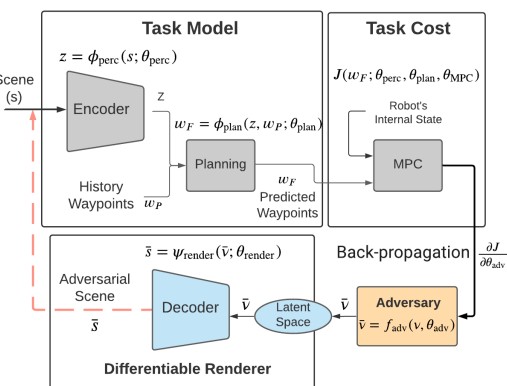

Figure 1: **Differentiable Robot Learning Architecture:** A robot observes an image $s$, which it maps through an encoder to yield scene embedding $z$. Given a history of waypoints $w_p$, a task module (planner) outputs a future set of waypoints $w_F$ to track. Finally, these waypoints are tracked using differentiable MPC that outputs a task cost $J$. We assume scenes $s$ are rendered by a generative model, such as a differentiable renderer (blue), that maps latent parameters describing a scene $v$ to an image $s$. **Our key contribution** is to train an adversary (orange) to generate synthetic, but realistic, images $\bar{s}$ that are poor for the current planner and controller. Re-training the planning module on these scenes improves robust performance on OoD data.

1. We develop a novel loss function that flexibly trades-off the realism of adversarial examples and their impact on control cost by learning perturbations in a latent visual space.
2. We present a method to automatically synthesize adversarial examples for vision-based MPC.
3. We show that re-training on these synthesized examples leads to better performance on unseen test data than today's standard task-agnostic data augmentation on diverse scenarios ranging from aerial navigation to arm manipulation in the photo-realistic RoboSuite simulator [28].

## 2   Problem Statement

We describe our problem using the information flow in our robot learning architecture (Fig. 1).

*Differentiable Perception and Planning Modules:*   The robot's perception module $\phi_{\text{perc.}}$ maps a high-dimensional visual observation $s$ into a low-dimensional scene embedding $z$, denoted by $z = \phi_{\text{perc.}}(s; \theta_{\text{perc.}})$. Here, $\theta_{\text{perc.}}$ are parameters of the perception module, such as a DNN. Next, the robot uses the scene embedding $z$ to plan a collision-free trajectory to achieve its task. Often, we can represent a robot's trajectory by a sequence of waypoints, such as a set of landmark poses in the $xy$ plane for a 2-D planar robot. Then, the differentiable planning module maps scene embedding $z$ and a vector of $P$ past waypoints $w_p$ to a vector of $F$ future waypoints to follow, denoted by $w_F = \phi_{\text{plan}}(z, w_p; \theta_{\text{plan}})$. Here, $\theta_{\text{plan}}$ are parameters of the differentiable planner. Henceforth, for brevity, we refer to the composition of the perception and planning modules as a task module $w_F = \phi_{\text{task}}(s, w_p; \theta_{\text{perc.}}, \theta_{\text{plan}})$, such as the cascaded operation of two DNNs.

*Differentiable MPC:*   Next, the robot must follow the waypoints $w_F$ with minimal control cost, while incorporating physical constraints. Suppose the robot is at some initial state $\zeta_t$ and has a planning horizon of $F$ steps. Then, its controller $\pi$ should map the initial state $\zeta_t$ and future waypoints $w_F$ into a sequence of controls from $t$ to $t+F$, denoted by $u_{t:t+F} = \pi(\zeta_t, w_F; \theta_{\text{MPC}})$. Here, $\theta_{\text{MPC}}$ could represent feedback gains for MPC. Crucially, we define a task (control) cost $J(w_F; \theta_{\text{perc.}}, \theta_{\text{plan}}, \theta_{\text{MPC}})$ which depends on the final waypoints the robot tracks $w_F$. Since the waypoints $w_F$ depend on the previous perception and planning modules, the cost $J$ depends on their parameters too. The MPC task cost $J$ could balance actuation effort and state deviations from a reference trajectory.

*Differentiable Rendering Module:* This paper analyzes how the ultimate MPC task cost $J$ would increase if the robot observes an adversarial visual input $\bar{s}$ instead of original image $s$. To do so, we need a process to generate, or render, realistic adversarial scenes $\bar{s}$. Fortunately, recent progress in generative models and differentiable graphics renderers makes this possible [15, 29]. A differentiable renderer $s = \psi_{\text{render}}(v; \theta_{\text{render}})$ generates the robot's scene $s$ from latent parameters $v$ that describe the scene and internal parameters $\theta_{\text{render}}$ of the rendering model. Often, the latent representation $v$ of scene $s$ could be interpretable and controllable, such as the pose, lighting, and texture of different objects. Without loss of generality, the renderer in our pipeline could also be the decoder in a Varational Autoencoder (VAE) [30] or a Generative Adversarial Network (GAN) [31]. All we require is a differentiable mapping that generates new scenes $\bar{s}$ from latent parameters $\bar{v}$.

*End-to-End Differentiable Architecture:* Since every block in Fig. 1 is differentiable, we have an end-to-end differentiable mapping from scene parameters $v$ to ultimate task cost $J$. Henceforth, we slightly re-define our notation for the task cost to be $J(v; \theta_{\text{perc.}}, \theta_{\text{plan}}, \theta_{\text{MPC}})$ to make it explicit that the cost depends on the latent scene representation $v$. This is because $v$ generates scene $s$, which is mapped to MPC waypoints $w_F$ through the task module. Thus, we can back-propagate the gradient of the MPC task cost $J$ with respect to scene parameters $v$ to efficiently search for adversarial $\bar{v}$.

*Train/Test Datasets:* We must first formalize our dataset notation. A dataset contains $N$ tuples of inputs $x$ and ground-truth labels $y$ denoted by $\mathscr{D} = \{x, y\}_{i=1}^{N}$. Specifically, each example $x = (s, v, w_P)$ represents the robot's scene input $s$, corresponding latent representation $v$ from the renderer, and waypoint history $w_P$. From these, the task model predicts a ground-truth target vector $y = w_F$ of future robot waypoints, which can come from an oracle planner or human demonstrations.

Henceforth, the subscript $b$ in the dataset notation $\mathscr{D}_b^a$ will represent the type of dataset, such as whether it is an original image, from data augmentation, or adversarial training. Likewise, the superscript $a$ will represent if the dataset is from the *train* or *test* distribution. For example, the original training and test datasets are given by $\mathscr{D}_{\text{orig}}^{\text{train}}$ and $\mathscr{D}_{\text{orig}}^{\text{test}}$. The training function TRAIN uses supervised learning to estimate correct waypoints $w_F$ given the scene $s$. The notation $\theta_{\text{perc.}}^0, \theta_{\text{plan}}^0 \leftarrow \text{TRAIN}(\mathscr{D}_{\text{orig}}^{\text{train}})$ indicates we train the nominal task model parameters on the original training dataset.

For a focused contribution, we consider a scenario where the robot has an MPC controller with fixed parameters $\theta_{\text{MPC}}$. Moreover, we have already pre-trained a nominal task model $\{\theta_{\text{perc.}}^0, \theta_{\text{plan}}^0\}$ on original images in $\mathscr{D}_{\text{orig}}^{\text{train}}$. Given these inputs, our problem is to generate an additional dataset $\mathscr{D}_{\text{new}} = \{x, y\}_{i=1}^{M}$ of $M$ datapoints such that re-training the task model on combined dataset $\mathscr{D}_{\text{orig}}^{\text{train}} \cup \mathscr{D}_{\text{new}}$ will minimize task cost $J$ on a held-out original $\mathscr{D}_{\text{orig}}^{\text{test}}$ and OoD test dataset $\mathscr{D}_{\text{OoD}}^{\text{test}}$. We only consider adding $M$ extra examples to limit training costs. Formally, our problem becomes:

**Problem 1 (Data Augmentation for Control)** *Given fixed MPC parameters $\theta_{\text{MPC}}$, original training dataset $\mathscr{D}_{\text{orig}}^{\text{train}}$, and a priori unknown test datasets, $\mathscr{D}^{\text{test}} = \mathscr{D}_{\text{orig}}^{\text{test}} \cup \mathscr{D}_{\text{OoD}}^{\text{test}}$, find a new dataset $\mathscr{D}_{\text{new}} = \{x, y\}_{i=1}^{M}$ of size $M$ such that:*

$$\mathscr{D}_{\text{new}}^* = \underset{\mathscr{D}_{\text{new}}}{\arg\min} \, \mathbb{E}_{(s', v', w_p, w_F') \sim \mathscr{D}^{\text{test}}} \, J(v'; \theta_{\text{perc.}}^*, \theta_{\text{plan}}^*, \theta_{\text{MPC}}),$$

$$\text{where} \ \ \theta_{\text{plan}}^*, \theta_{\text{perc.}}^* \leftarrow \text{TRAIN}(\mathscr{D}_{\text{orig}}^{\text{train}} \cup \mathscr{D}_{\text{new}}).$$

It is hard to analytically find the best dataset of $M$ points to minimize task cost on OoD data, since this requires re-training non-convex DNNs. We now describe our adversarial training approach.

## 3 Adversarial Scenario Generation Algorithm

We aim to generate a dataset of adversarial scenarios $\mathscr{D}_{\text{adv}}$ where the original task model performs poorly. Then, by re-training the task model on the joint dataset $\mathscr{D}_{\text{orig}}^{\text{train}} \cup \mathscr{D}_{\text{adv}}$, we hope to improve its robust generalization. To do so, we introduce a differentiable adversary to generate the new dataset $\mathscr{D}_{\text{adv}}$, shown in orange in Fig. 1. The adversary maps the original latent scene representation $v$ to a perturbed representation $\bar{v} = f_{\text{adv}}(v; \theta_{\text{adv}})$, where $\theta_{\text{adv}}$ are trainable parameters.

*Adversary Loss Function:* We now introduce a novel loss function to train the adversary $f_{\text{adv}}$. Intuitively, the adversary's loss function must perturb the latent scene representation $v$ to $\bar{v}$ such that the

task cost increases. At the same time, it must ensure that the new rendered scene from $\bar{v}$ is realistic enough to plausibly encounter during real-world testing. Therefore, the following adversarial loss function delicately balances the task cost and distance loss between $\bar{v}$ and $v$.

**1 Input:** $\theta_{\text{perc.}}^0, \theta_{\text{plan}}^0, \theta_{\text{MPC}}, \mathscr{D}_{\text{orig}}^{\text{train}}$
**2** Initialize empty new adversarial dataset $\mathscr{D}_{\text{adv}} = \{\}$
**3 for** $(x_i = \{s_i, v_i, w_P\}, y_i) \sim \mathscr{D}_{\text{orig}}^{\text{train}}, 0 \leq i \leq M$ **do**
**4** $\quad$ Init. Adv. $f_{\text{adv}}(v_i; \theta_{\text{adv}}^0)$ with random $\theta_{\text{adv}}^0$
**5** $\quad$ Init. Adv. latent rep. $\bar{v}^0 = v_i$
**6** $\quad$ Best Task Loss $J^\star = -\infty$
**7** $\quad$ Init. Adv. tuple $(\bar{x}, \bar{y}) = (x_i, y_i)$
**8** $\quad$ **for** $k \leftarrow 1$ **to** $K$ **do**
**9** $\quad\quad$ Get Adv. latent params
$\quad\quad\quad \bar{v}^k = f_{\text{adv}}(\bar{v}^{k-1}; \theta_{\text{adv}}^{k-1})$
**10** $\quad\quad$ Render scene $\bar{s}^k = \psi_{\text{render}}(\bar{v}^k; \theta_{\text{render}})$
**11** $\quad\quad$ Calculate task cost $J(\bar{v}^k; \theta_{\text{adv}}^{k-1})$
**12** $\quad\quad$ Calc. Loss $L = \mathscr{L}(v_i, \bar{v}^k; \theta_{\text{adv}}^{k-1})$
**13** $\quad\quad$ $\theta_{\text{adv}}^k \leftarrow \text{BackProp}(\theta_{\text{adv}}^{k-1}, L)$
**14** $\quad\quad$ **if** $J(\bar{v}^k, \theta_{\text{adv}}^{k-1}) \geq J^\star$ **then**
**15** $\quad\quad\quad$ Update most adv. scene
$\quad\quad\quad\quad \bar{x}^k = \{\bar{s}^k, \bar{v}^k, w_P\}$
**16** $\quad\quad\quad$ Update adv. example $(\bar{x}, \bar{y}) = (\bar{x}^k, y_i)^a$
**17** $\quad\quad\quad$ $J^\star \leftarrow J$
**18** $\quad\quad$ **end**
**19** $\quad$ **end**
**20** $\quad$ Append adv. example $(\bar{x}, \bar{y})$ to $\mathscr{D}_{\text{adv}}$
**21 end**
**22 Return:** $\theta_{\text{perc.}}', \theta_{\text{plan}}' \leftarrow \text{Train}(\mathscr{D}_{\text{orig}}^{\text{train}} \cup \mathscr{D}_{\text{adv}})$
**Algorithm 1: Adversary Training**

---

[a] As described later in Sec. 5.7, our method is general, and we can either set the label for an adversarial image to be the same label as the original image or an altogether new label. This choice depends on how much the difference between the adversarial scene and the original scene affects task completion and semantic meaning. For example, if an adversarial trajectory changes the original path significantly in motion planning, we can generate new waypoint labels by invoking an offline oracle planner. Suppose we aim to slightly distort an image to emulate different weather and background scenarios, but the target prediction remains the same. In that case, we can use the same label as the original image.

*Adversarial Task Cost:* Given an original latent scene representation $v$, the adversary generates an adversarial representation by $\bar{v} = f_{\text{adv}}(v; \theta_{\text{adv}})$. Then, we simply invoke the end-to-end differentiable modules in Fig. 1. Importantly, we use the pre-trained task module parameters $\{\theta_{\text{perc.}}^0, \theta_{\text{plan}}^0\}$ since they will only be re-trained after the adversarial dataset generation procedure. We then calculate the new task cost $J(\bar{v}; \theta_{\text{perc.}}^0, \theta_{\text{plan}}^0, \theta_{\text{MPC}}, \theta_{\text{adv}})$. Importantly, the task cost $J$ depends on the adversary parameters $\theta_{\text{adv}}$ since they generate perturbed scene representation $\bar{v}$. For a concise notation, we will represent the task cost $J(\bar{v}; \theta_{\text{perc.}}^0, \theta_{\text{plan}}^0, \theta_{\text{MPC}}, \theta_{\text{adv}})$ as $J(\bar{v}; \theta_{\text{adv}})$, since the parameters $\theta_{\text{perc.}}^0, \theta_{\text{plan}}^0, \theta_{\text{MPC}}$ are fixed.

*Consistency (Distance) Loss:* To promote generation of plausible adversarial images, we introduce a consistency loss $I(v, \bar{v}) \in \mathbb{R}$. Calculating the distance loss in the *latent space* is a novelty of our loss function based on the observation that similar scenes $s$ and $\bar{s}$ might be very far in the high-dimensional image space due to task-irrelevant pixel variation but might be close in the task-relevant latent space of the renderer $\psi_{\text{render}}$. In our experiments, we use the $L_2$ norm distance between $\bar{v}$ and $v$, although our method can accomodate any differentiable distance metric.

*Overall Adversarial Loss Function:* Finally, our adversarial loss function trades off extra control cost while incentivizing scene realism. During adversarial training, only the adversary's parameters $\theta_{\text{adv}}$ are trained while the initial task module parameters are fixed. The distance loss is weighted by $\kappa$, which can be flexibly set by a roboticist depending on how adversarial they prefer rendered scenarios to be. The loss to minimize is:

$$\mathscr{L}(v, \bar{v}; \theta_{\text{adv}}) = -J(\bar{v}; \theta_{\text{adv}}) + \kappa I(v, \bar{v}). \tag{1}$$

*Adversary Training:* Algorithm 1 shows our complete training procedure. The inputs are an original training dataset $\mathscr{D}_{\text{orig}}^{\text{train}}$, fixed MPC parameters $\theta_{\text{MPC}}$, and pre-trained task module parameters $\theta_{\text{perc.}}^0, \theta_{\text{plan}}^0$. We randomly sample $M$ datapoints from the training distribution and generate their corresponding adversary by performing gradient descent on our loss function (Eq. 1). Then, we save the most adversarial synthetic examples for subsequent data augmentation. To do so, we use a new adversary for each datapoint rather than training a common adversary for the whole dataset.

On line 3, we iterate through $M$ random samples of the original dataset $\mathscr{D}_{\text{orig}}^{\text{train}}$. Each datapoint $x_i$ is a tuple of input scene $s_i$, corresponding latent representation $v_i$, and past waypoints $w_P$. Lines 4-7 describe how we initialize a *new trainable* adversary for each datapoint. In line 4, we initialize a new adversary $f_{\text{adv}}$ with random weights $\theta_{\text{adv}}^0$, which will be specific to the datapoint $x_i$. Since we train the adversary parameters for $K$ gradient descent steps, we use superscript $k$ to indicate the

adversary's parameters at training step $k$. Likewise, we index the new adversarially-generated latent representation at each step $k$ by $\bar{v}^k$. Finally, in lines 5-7, we initialize a tuple $(\bar{x}, \bar{y})$, which will indicate the adversarially generated datapoint and label, with the *original* training dataset's values.

The crux of our algorithm is in lines 8-18, where we perform $K$ gradient update steps to train the adversary $f_{\text{adv}}$. In each gradient step $k$, we update the adversary parameters to $\theta_{\text{adv}}^k$ using loss function $\mathscr{L}$ and back-propagation (lines 9-13). In lines 15-19, we store the adversarial tuple $(\bar{x}^k, y_i)$ if it has higher task cost $J$ than the previous best task cost $J^\star$. Finally, after $K$ gradient steps, we append the best adversarial datapoint $(\bar{x}, \bar{y})$ to dataset $\mathscr{D}_{\text{adv}}$ in line 20. Crucially, we only train the adversary parameters $\theta_{\text{adv}}$ during dataset generation. At the end, we have generated a new adversarial dataset, which is augmented with the original dataset to re-train the task model parameters $\theta'_{\text{perc.}}, \theta'_{\text{plan}}$ on line 22. Our method's runtime is linear with the size of the dataset but is completely parallelizable since we can train a new adversary for each datapoint in parallel during the *for* loop in line 3.

## 4  Experimental Results

We now evaluate our method (Alg. 1) on three diverse tasks. The first task is a toy example of 2-D robotic motion planning. The second task concerns manipulation in the RoboSuite [28] environment. Finally, in the third experiment, we make an autonomous aircraft taxi by tracking a runway center-line based on visual inputs in challenging weather conditions.

*Adversary Model:* We use a linear model for our adversary $f_{\text{adv}}(v; \theta_{\text{adv}}) = \theta_{\text{adv}} v$, where $\theta_{\text{adv}} \in \mathbb{R}^{n \times n}$ and $v \in \mathbb{R}^n$, which was expressive enough to generate realistic scenarios that are poor for control. Our system can easily use more complex DNN adversary models since it is fully differentiable.

*Distance Loss and Regularization:* We use the $L_2$-norm as the consistency loss $\text{I}(v, \bar{v}) = ||v - \bar{v}||_2^2$ in Eq. (1). As shown in the experiments, the consistency loss weight $\kappa$ can be flexibly set by the user based on how adversarial they want synthetic examples to be.

*Variational Autoencoder (VAE):* We use a VAE [30] as the differentiable renderer $\psi_{\text{render}}$ from Sec. 2, because of their stable training procedures compared to GANs. Notably, we only use the VAE *Decoder* as the rendering module, which takes parameters $v$ as input and outputs the image $s$ of the scene. We keep the *Decoder* frozen after training it on the training datasets.

*Differentiable MPC:* All tasks use the task cost $J(w_F)$ for an MPC problem with quadratic costs and linearized dynamics constraints, as shown in Appendix 5.3. Our key step is to calculate the sensitivity of the task cost $J$ to adversarial perturbations in the image $s$ in order to train the adversary. To do so, we use the PyTorch CVXPYLAYERS library [32] to calculate the gradient of MPC's solution and task cost (a convex program) with respect to its problem parameters (i.e., waypoint predictions $w_F$). Then, we back-propagate through the differentiable pipeline in Fig. 1 to compute the gradient of MPC's task cost with respect to the adversary's parameters.

*Architectures:* All experiments use DNNs for all modules in Fig. 1 except model-based control and the adversary. The perception module $\phi_{\text{perc.}}$ is a convolutional VAE *Encoder* that maps image $s$ to an embedding $z$ used by the planner. For the planner, we achieved success with simple multi-layer perceptrons (MLPs) that output waypoints, such as locations for the arm to reach in manipulation or desired poses for the aircraft. Our algorithm is agnostic to the type of deep network since it aims to improve any general, differentiable task model. Further training/model details are in Appendix 5.4.

**Datasets and Benchmarks:**  We compare task models trained on the following datasets:

1. ORIGINAL: The task model (perception and planning model) is *only* trained on $\mathscr{D}_{\text{orig}}^{\text{train}}$.
2. DATA ADDED: We add more training examples from the same (or very similar) distribution as the original training dataset, denoted by $\mathscr{D}_{\text{add}}^{\text{train}}$. This tests whether more examples are necessary to achieve better performance. The task model is then trained on the union of the original and added dataset, denoted by $\mathscr{D}_{\text{add}}^{\text{train}} \cup \mathscr{D}_{\text{orig}}^{\text{train}}$.
3. DATA AUGMENTATION: We apply standard *task-agnostic* data augmentation to the original training data. For image inputs $s$, we applied random contrast, random brightness, and random blur. The augmented dataset is denoted by $\mathscr{D}_{\text{aug}}^{\text{train}}$ and we re-train the task model on $\mathscr{D}_{\text{aug}}^{\text{train}} \cup \mathscr{D}_{\text{orig}}^{\text{train}}$.
4. CURL: CURL[33] uses contrastive learning and standard data augmentation techniques for targeted feature extraction, leading to faster training and robust feature learning (Sec. 5.6).

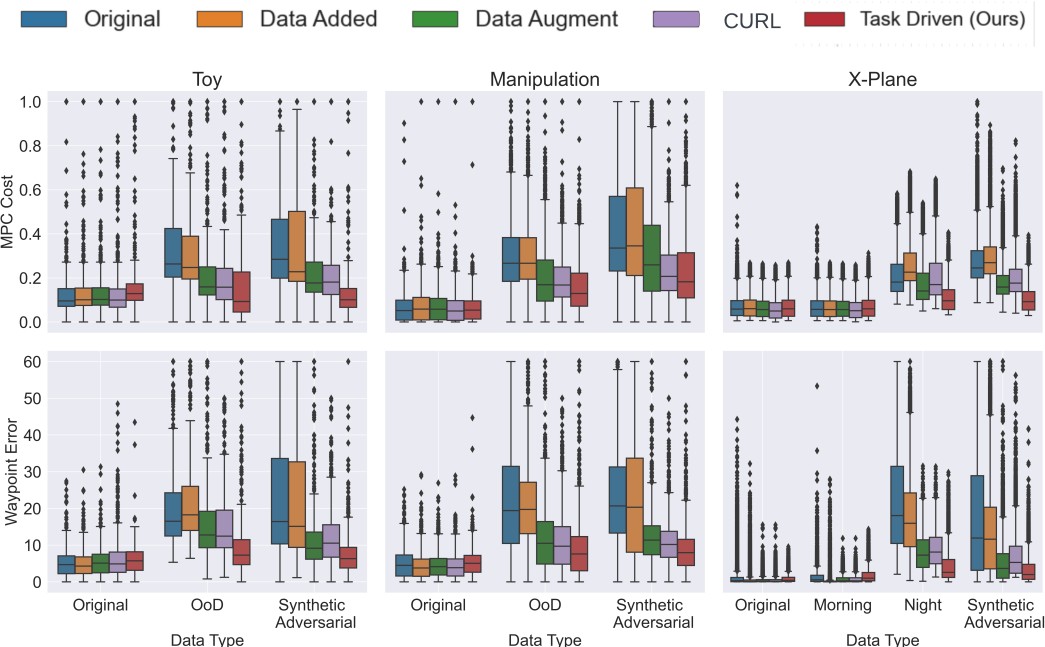

Figure 2: **Benefits of Task-Driven Data Augmentation for Simple 2D Motion Planning, Robot Manipulation and X-Plane Dataset:** We show the MPC task cost $J$ (top) and MSE for waypoint predictions (bottom) for all training schemes on held-out test environments. The x-axis shows different test conditions, and lower task cost and waypoint MSE (y-axis) are better. Our TASK DRIVEN scheme (red) works on par with other task models on the *Original* test data for both metrics and experiments. However, it significantly outperforms other task models on challenging scenarios in the *out-of-distribution (OoD)* dataset (*Night* images for X-Plane), and *Synthetic Adversarial* test datasets. We beat the baselines of task-agnostic data augmentation (green) and CURL (purple), which often apply image augmentations that are ultimately benign for MPC. The DATA ADDED training has high variance on OoD and Adversarial datasets since it overfits to the Original dataset.

5. TASK DRIVEN (Ours): We use Alg. 1 to synthesize adversarial dataset $\mathscr{D}_{\text{adv}}^{\text{train}}$ and then re-train the task model on $\mathscr{D}_{\text{adv}}^{\text{train}} \cup \mathscr{D}_{\text{orig}}^{\text{train}}$.

**Diverse Robotic Tasks:** We visualize all model-based robotic tasks in Fig. 4. We use MPC with quadratic costs and linearized robot dynamics in all the experiments. We use linear obstacle avoidance constraints, calculated by drawing perpendicular hyperplanes from the obstacles, discussed further in Appendix 5.3. **Toy 2-D Planning:** In this task, a robot must navigate around obstacles (large ellipsoids) to reach a goal location (green star) using only images $s$ of the 2-D workspace. The planning module learns to output waypoints $w_F$ that mimic a Frenet Planner [34], which are tracked by MPC. The renderer parameters $v$ control the size and location of obstacles in image $s$. The OoD test dataset consists of larger, more varied obstacles with a different distribution than those in the training dataset. **RoboSuite Manipulation:** A Franka Emika Panda arm [35] needs to pick up a blue ball while avoiding red boxes. The robot observes images $s$ from an overhead camera, and the planning module outputs a collision-free trajectory with waypoints $w_F$. Then, MPC tracks the trajectory in task-space using the RoboSuite joint controller [35]. The OoD test dataset consists of larger red boxes randomly placed in the scenario compared to those in the training dataset. **X-Plane Aerial Navigation:** This scenario is inspired by academic work for the DARPA Assured Autonomy program [36–39]. An airplane passes images $s$ from a wing-mounted camera into a perception model that estimates its distance and heading angle relative to the center-line of a runway. Then, it uses differentiable MPC to track waypoints $w_F$ that approach the runway center-line. We use a standard benchmark dataset for robust perception [37–39] from the photo-realistic X-Plane simulator [40], consisting of images from the plane's camera in diverse weather conditions and aircraft poses. Importantly, we test whether a perception model trained on only bright afternoon/morning conditions generalizes to OoD night test scenarios. We also used the Frenet Planner [34] to generate ground truth waypoints of the adversarially synthesized scenarios for Toy 2-D Planning and RoboSuite.

We first trained a VAE on all three domains on all the datasets. Next, we trained the task model (perception and planning models) for a subset of training data defined in Sec. 4. Due to space limits,

we provide further implementation details in Appendix 5.4 and the MPC task cost in Eq. 2. We now evaluate the performance of all training schemes on OoD test datasets. All the results presented are run for 5 random seeds and between $2.5 - 10$k test images per dataset (see Appendix).

| Domain | Dataset | Avg MPC Cost Difference (%) | Avg Waypoint Error Difference (%) |
|---|---|---|---|
| Toy | $\mathscr{D}_{\text{OoD}}^{\text{test}}$ | 33.6% | 66.1% |
| | $\mathscr{D}_{\text{adv}}^{\text{test}}$ | 49.9% | 51.5% |
| Manipulation | $\mathscr{D}_{\text{OoD}}^{\text{test}}$ | 29.1% | 37.1% |
| | $\mathscr{D}_{\text{adv}}^{\text{test}}$ | 36.2% | 32.8% |
| X-plane | $\mathscr{D}_{\text{night}}^{\text{test}}$ | 36.2% | 89.2% |
| | $\mathscr{D}_{\text{adv}}^{\text{test}}$ | 41.7% | 69.1% |

Figure 3: **TASK-DRIVEN vs DATA AUGMENTATION schemes for MPC Costs and Waypoint Error**: We compare the percentage improvement in the Avg. MPC cost and Avg. Waypoint Error for the test OoD and synthetic adversarial test datasets. On all domains, our TASK-DRIVEN scheme performs significantly better than DATA AUGMENTATION.

**Results:** Our experiments show that training the task model with additional adversarial data using Alg. 1 reduces the task cost $J$ and the mean-squared error (MSE) for waypoint prediction compared to all benchmarks. Finally, we visualize how our rendered scenarios are adversarial and realistic.

*Quantitative Results:* Fig. 2 (top) shows the MPC task cost for all training schemes on test datasets. On the original test dataset, our TASK-DRIVEN scheme (red) performs on par with the ORIGINAL (blue) scheme and slightly worse than the DATA-ADDED (orange), DATA AUGMENTATION (green) and CURL (purple) schemes. The DATA-ADDED scheme's performance is expected because it is trained on more original data than our TASK-DRIVEN scheme, which allows it to perform slightly better on the original test dataset. However, the key benefits of our approach are shown on the synthetic adversarial test dataset and held-out OoD dataset (real X-Plane night images). We run Alg. 1 on the held-out original test dataset to generate unseen adversarial images which form the synthetic adversarial test dataset. As shown in Fig. 2 (bottom), these benefits arise because our TASK-DRIVEN scheme achieves lower waypoint prediction MSE than competing benchmarks. While all methods perform similarly on the original test dataset, the key gains of our scheme are on the adversarial test dataset and held-out OoD dataset (night images for X-Plane). We compare our TASK-DRIVEN scheme (red) with DATA AUGMENTATION in Table 3, and Table 6 shows the cost differences are statistically significant with a Wilcoxon p-value less than $10^{-4}$. Even though the CURL scheme performs best on the original test dataset, it fails to generalize to the test OoD dataset. This is not surprising even though it learns comparably better on the original training dataset; nothing is helping CURL generalize better to the OoD dataset. Additionally, CURL's performance is similar to DATA AUGMENTATION, since both these methods are exposed to similar augmentation during training.

*Qualitative Results:* Fig. 4 (bottom row) shows a challenging adversarial scenario that is automatically synthesized by our method in dataset $\mathscr{D}_{\text{adv}}^{\text{train}}$ for the corresponding original training example from $\mathscr{D}_{\text{orig}}^{\text{train}}$ (top row). For the toy planning scenario and manipulation task, our training scheme brings obstacles closer to the robot's originally intended waypoints and path, making it harder to find a collision-free path. Naturally, this increases the task cost since the robot has to make wider turns and swerves. Specifically, for the manipulation task, the adversary moves red obstacles around the blue target to make it harder to reach and decreases the target size. For the X-Plane scenario, our training scheme learns to blur the runway center-line to emulate a foggy condition, leading to higher waypoint prediction error and task cost. Therefore, rather than augmenting our dataset with well-understood training scenarios, our scheme targetedly adds challenging scenarios that make the model more robust. Finally, Fig. 5 shows the relative distance between different test datasets' latent representation in the X-Plane renderer when projected in a 2D space using the t-SNE method [41]. Interestingly, our task-driven scheme can add synthetic data points (yellow) that emulate and anticipate the OoD night distribution. However, points added by standard data augmentation (pink) do not overlap much with the night data (green), explaining why the benchmark generalizes poorly.

**Limitations:** We note that (only for the X-Plane experiment), the original VAE renderer was trained on the whole training dataset. For example, the renderer for the X-Plane experiment was trained on night training images so that the VAE could render a night scene in the first place. However, the initial perception and planning models were never exposed to a single night (OoD) image and were strictly trained on only the original afternoon training data. This scenario is practically motivated. Consider a state-of-the-art differentiable renderer that can map a continuum of scene parameters $v$ to scenes $s$.

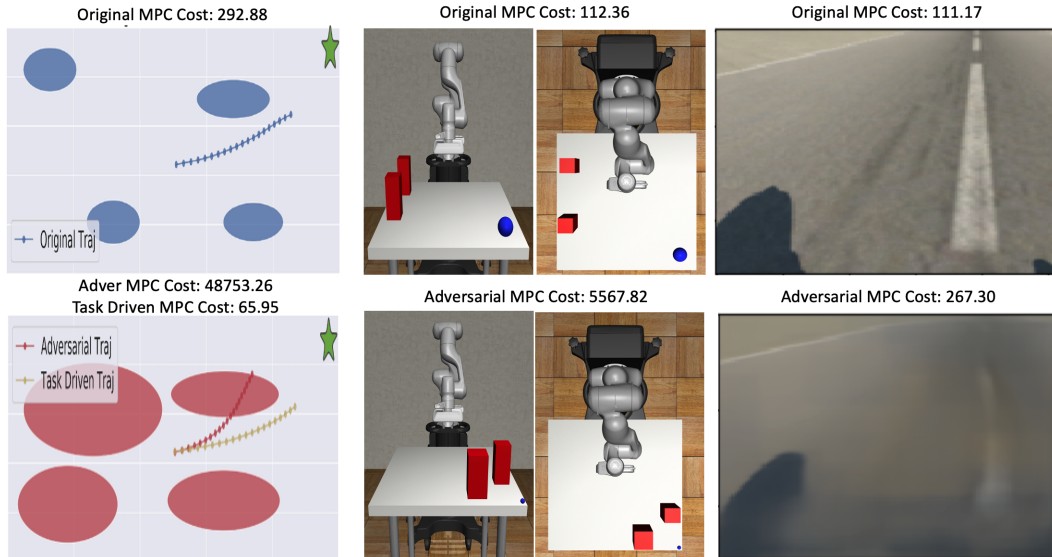

Figure 4: **Synthesized Adversarial Scenarios:** The top row contains images from the original dataset with the corresponding MPC task cost if the predicted waypoints were correctly followed. The bottom row shows the corresponding adversarial scenario generated by our method. As expected, they lead to higher MPC costs by introducing obstacles near a goal or obfuscating the runway center-line. Figs. 11-13 show more examples.

An infinite number of possible images can be generated from the continuous latent space. However, due to compute limits, we can only train the task model on a finite set of scenes $v$ from a specific training distribution. Thus, we can use our method to automatically and efficiently find new scenes to maximally improve the perception model. Crucially, we must be able to render these new scenes, which is possible with a differentiable renderer. Since we did not have a differentiable renderer for the X-Plane experiment, we used a VAE and ensured it could generate an OoD night image via pre-training for the night class. Additionally, for the cases where the same labels as training cannot be used (Toy Example and Manipulation), our method relies on an offline planner to label the adversarial scenarios. We discuss the labeling procedure further in Sec. 5.7.

**Conclusion:** This paper presents a principled method to automatically synthesize adversarial scenarios tailored to model-based robotic control. Our key contribution is to compute the sensitivity of MPC to perception errors, which

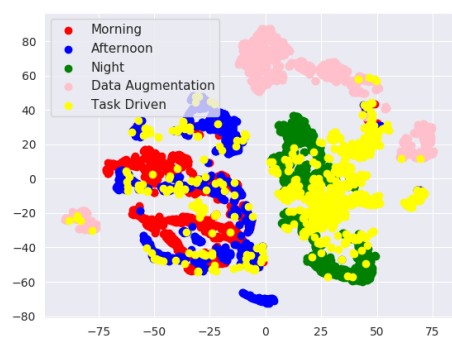

Figure 5: **X-Plane VAE Latent Space Visualization:** We visualize the latent representation $z = \phi_{\text{perc.}}(s; \theta_{\text{perc.}})$ for each test dataset of scenes $s$ after performing dimensionality reduction using t-SNE [41]. The scenes generated by our task-driven method (yellow) are very close to the OoD test night dataset (green), which explains why it generalizes so well. However, the scenes from standard data augmentation (pink) are far from the OoD night scenes, thus explaining why it does not yield generalizable models for control.

in turn guides better synthesis of adversarial scenarios and robust generalization in state-of-the-art robotic simulators. Our method can benefit applications like self-driving cars by automatically synthesizing rare, OoD scenarios before costly testing and data collection on physical platforms. In future work, we plan to provide theoretical guarantees for an illustrative example with a linear adversary, simple linear renderer/scene generator, and linear MPC control problem. Finally, we plan to test the generalization ability of our approach on a real robot operating in OoD lighting, weather, and terrain conditions.

**Acknowledgements:** This material is based upon work supported in part by the Office of Naval Research (ONR) under Grant No. N000142212254. We also gratefully acknowledge the support of the Lockheed Martin AI Center for this research under contract number RPP015-MRA16-005. Any opinions, findings, conclusions, or recommendations expressed in this material are those of the author(s). They do not necessarily reflect the views of ONR or the Lockheed Martin AI Center.

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
