# OpenReview forum: "Synthesizing Adversarial Visual Scenarios for Model-Based Robotic Control"
_robot-learning.org/CoRL/2022/Conference — CoRL 2022 Poster_

### Official Review · Reviewer_bmgX · 2022-07-28

**Originality:** Good
**Technical Quality:** Very Good
**Clarity Of Presentation:** Very Good
**Impact:** 3

**Recommendation:**

Weak Accept: I recommend accepting the paper, but will not argue for my recommendation if the majority of other reviewers have a different opinion.

**Summary:**

This paper presents a method to generate adversarial visual scenarios for robotic tasks by reasoning about the impact of visual changes in the scene on the control cost. This is achieved by establishing an end-to-end differentiable autonomy stack. The perception and planning modules are re-trained with an augmented dataset that contains the original dataset along with the adversarial scenes. The re-trained autonomy stack demonstrates significant improvements on out-of-distribution scenarios.

**Issues:**

Nothing major here. Most weaknesses I mentioned above are difficult to address within the rebuttal period and are more likely to be extensions or follow-up studies for this paper. The two main points I would like to be addressed are:
1. Improved limitations section.
2. The text in lines 172-174 seems to have much lower line spacing compared to the rest of the text.

**Quality Of The Limitations Section:**

Limitations are not well addressed

**Reviewer Expertise:**

3: The reviewer is fairly confident that the evaluation is correct

**Robotics Focus:**

Highly relevant to robotics but no hardware experiments

**Strengths And Weaknesses:**

Strengths:
1. The task-driven adversarial approach automatically finds the perturbations that are relevant to the robot’s task as encoded by the cost function. Although using the planning and control cost for adversarial generation is not an entirely novel idea (see A. Z. Ren, A. Majumdar. "Distributionally robust policy learning via adversarial environment generation." for instance), I found the execution to be refreshing. Instead of relying on an end-to-end neural network controller, the paper resorts to the use of differentiable MPC which is a lot more common in practical autonomy stacks than end-to-end neural network controllers.
2. End-to-end differentiability of the approach allows for easier training and adversarial scene generation.
3. Experiments are reasonably thorough.

Weaknesses:
1. This framework may not be easily generalizable to different autonomy stacks. For instance, if the stack employs a sampling-based planner or a trajectory prediction module, retaining the end-to-end differentiability of the approach will be fairly challenging.
2. A benefit of training task-agnostic perception networks is disentangling their dependence on the task itself, and therefore promoting overall generalizability to other tasks with the same perception network. It seems that specializing the perception system to a single task could adversely affect its performance on other tasks.
3. Simply requiring the adversarial scenario to be near the actual scenario seems like a na\”ive way of enforcing realism. As an example, consider a manipulator picking a block from a table. If the adversarial scenario has a mesh of the block at the exact same location as in the real scenario, but the object mesh penetrates into the mesh of the table by a tiny amount, this scenario would be adversarial and in close proximity to the true scenario in terms of distance between visual features; however, clearly it is not actually realistic.
4. Limitations are not very insightful. I had a hard time following what that section was saying.


**Summary Of Recommendation:**

I recommend a Weak Accept. The strengths of the paper outweigh the weaknesses I listed above.

---

### Official Review · Reviewer_rbua · 2022-07-31

**Originality:** Fair
**Technical Quality:** Fair
**Clarity Of Presentation:** Fair
**Impact:** 3

**Recommendation:**

Weak Accept: I recommend accepting the paper, but will not argue for my recommendation if the majority of other reviewers have a different opinion.

**Summary:**

The paper aims to automatically synthesize adversarial visual scenarios for multi-step, model-based control tasks in order to re-train more robust robotic perception models.  While existing work focuses on synthesizing adversarial examples for single-step vision tasks, this paper instead looks to synthesizing adversarial examples for multi-step, model-based control. Using adversarial training and a differentiable MPC module, the paper is able to improve control performance on out-of distribution scenarios when compared to naive data augmentation strategies. This is validated on 3 domains: (1) navigation, (2) manipulation in RoboSuite, (3) drone control. All experiments are done in simulation.

 The main contribution of the work is automatically synthesize adversarial examples for vision-based MPC via a loss function that trades-off the realism of adversarial examples and their impact on a control cost.

**Issues:**

See weaknesses.

**Quality Of The Limitations Section:**

Additional details required

**Reviewer Expertise:**

4: The reviewer is confident but not absolutely certain that the evaluation is correct

**Robotics Focus:**

Relevant but unlikely to deploy to hardware in near future

**Strengths And Weaknesses:**

**Strengths**

- The idea of synthesizing realistic scenes from latents that are poor for control, but consistent with naturally-occurring, is interesting.
- The work does a good job distinguishing itself from robust adversarial reinforcement learning (RARL), Fast Gradient Sign Method (FGSM), and gradient-free Bayesian Optimization.
- The work evaluated across a broad set of domains.
- The proposed Adversary Loss Function is motivated and described well.
- The different datasets used (Original, Data Added, Data Augmentation) are sufficient for a good ablation study.

**Weaknesses**

- I disagree with the statement that data augmentation, domain randomization, etc, "improve the vision model in isolation". Many works do end-to-end control (image-to-action using augmentation and/or domain randomization [a, b, c]; in these works, vision is not separated from the control, and so the augmentation and domain randomization effects both the vision and control.  Perhaps this can be discussed?
- All 3 domains are very visually simple scenes, and so rendering adversarial examples are easy; unstructured real-world scenes would be much harder, and this method could fail in these domains. Could the authors elaborate on how they think this method can scale?
- I was excited to get to the results section while reading the paper, but was disappointed to see the very little difference between the proposed method and the simple data augmentation baseline in Fig 2. It begs the question: is the very small performance gain worth it given that you have to spend extra time generating the adversarial dataset?
- Fig/table 3 was confusing to me, and somewhat uninformative. Isn’t this the same information that is shown in Fig 2, but just in percentage points form?
- The paper does not make it clear at all that the renderer was trained on both the train and OoD test set(!). In line 214, the paper states that “We keep the Decoder frozen after training it initially on the whole dataset”; this does not imply that we train on the held-out OoD data… It is only until the limitations section that the paper states: “… For example, the renderer for the X-Plane experiment was trained on night training images so that the VAE could render a night scene in the first place” — where ‘’night’ scenes are considered OoD. This should be made clear in line 214. Moreover, if this is indeed the case, doesn’t it go against the whole purpose of the method? If we have the OoD to train the renderer, then why don’t we also include it in the perception and planning dataset?

- No real-world experiments. Given that this work is only in simulation, a discussion on how this work could be extended to the real world would be very welcome.

**Other minor comments / questions**

- The RoboSuite experiments looks like you have 2 camera images that make up the state; how do you ensure consistency between these 2 views when rendering adversarial examples?  -I would suggest removing “photo-realistic” from “photo-realistic RoboSuite”; it’s subjective, and in my view, RoboSuite is far from photo-realistic.

- A discussion on the cost of generating the final combined dataset would be helpful. Also an analysis on M (the number of rounds we do for building the adversarial dataset would be helpful).

- Rephrase Fig2 caption: “We beat today’s method …”

[a] James et al. "Transferring end-to-end visuomotor control from simulation to real world for a multi-stage task." Conference on Robot Learning. PMLR, 2017.

[b] Kalashnikov et al. "Scalable deep reinforcement learning for vision-based robotic manipulation." Conference on Robot Learning. PMLR, 2018.

[c] Matas et al. "Sim-to-real reinforcement learning for deformable object manipulation." Conference on Robot Learning. PMLR, 2018.

**Summary Of Recommendation:**

I am currently on the fence with the paper, and have aired on the side of caution. A strong rebuttal could increase my rating. My 2 main point of concern are (1) the small performance gain vs. the increased method complexity, and (2) the worry that this wont scale to more unstructured environments.

---

### Official Review · Reviewer_bY11 · 2022-08-01

**Originality:** Good
**Technical Quality:** Good
**Clarity Of Presentation:** Good
**Impact:** 3

**Recommendation:**

Weak Accept: I recommend accepting the paper, but will not argue for my recommendation if the majority of other reviewers have a different opinion.

**Summary:**

This work, creates additional datasets of adversarial examples tailored to multi-step model based control. This work uses differentiable MPC methods to calculate the sensitivity of a controller to state estimation errors. The models are then retrained on the augmented datasets to improve performance in out of distribution settings.





**Issues:**

Weak baselines. CURL would be a stronger comparison.

**Quality Of The Limitations Section:**

Limitations are addressed clearly

**Reviewer Expertise:**

3: The reviewer is fairly confident that the evaluation is correct

**Robotics Focus:**

Highly relevant to robotics but no hardware experiments

**Strengths And Weaknesses:**

Strengths:
- Compelling to explicitly leverage the MPC task cost to target pixels most salient for control.

Weakness:
From CURL (https://arxiv.org/pdf/2004.04136.pdf)
'''
Random crop data augmentation has been crucial for the performance of deep learning based computer vision systems in
object recognition, detection and segmentation (Krizhevsky et al., 2012; Szegedy et al., 2015; Cubuk et al., 2019; Chen et al., 2020).
'''
Random image cropping was also used in Mt-OPT (https://arxiv.org/abs/2104.08212)

I worry that the data augmentation baseline in the present work of random contrast, brightness and blur is not actually a rich enough set of augmentations.  It maybe useful to grab an augmentation method from a specific past work. I think the augmentation from the CURL work would be particularly relevant, as would a comparison to CURL in general.



**Summary Of Recommendation:**

Overall paper is well written, and the idea plays well with the advances in image generation. I would really like to see stronger comparisons to other augmentation approaches. Particularly CURL given that it is recent and open source.

---

### Official Review · Reviewer_2DMp · 2022-08-04

**Originality:** Very Good
**Technical Quality:** Good
**Clarity Of Presentation:** Very Good
**Impact:** 4

**Recommendation:**

Weak Reject: I recommend rejecting the paper, but will not argue for my recommendation if the majority of other reviewers have a different opinion.

**Summary:**

The authors present a data augementation that is task driven. Their approach leverages the output of the model predictive controller to generate adversarial scenarios that are use to retrain the perceptual model. They show that this approach is more efficient than task agnostic data augmentation.

**Issues:**

see above

**Quality Of The Limitations Section:**

Additional details required

**Reviewer Expertise:**

4: The reviewer is confident but not absolutely certain that the evaluation is correct

**Robotics Focus:**

Highly relevant to robotics but no hardware experiments

**Strengths And Weaknesses:**

Strrengths:
 - The problem is very well-articulated and motivated.
 - The idea is simple
 - The method is demonstrated on a diversity of robot tasks and shows improvement.

Major Weaknesses / Questions:
 - If I understand correctly, the *label* for the adversarial cases (i.e., the ground truth future waypoints) are the same as the sampled data points that were used to generate them. This can be seen from Line 16 in Algorithm 1. I don't see how this assumption can be valid in general. For example if we consider Figure 4, in both the left and middle cases the future waypoints would have to change in order to achieve the task in the adversarial scenarios. I would like the authors to justify this choice

 - I am confused about the latent space $v$. On Line 169-170 it is stated that "if the the latent space parameters $v$ have physical meaning, such as the pose of key objects" but in your experiments you generate this latent space $v$ with a VAE model. So in what settings is the latent space interpretable?

 - In the evaluation you use the "Waypoint Error". To what degree are the ground truth waypoints actually unique though. Imagine the case where the robot should move forward in a straight line (the 1D case). Waypoints at any point along this line should be considered to be optimal. In general I would like more detail about this metric, including how these ground truth waypoints are generated and to what degree you can guarantee that they are unique.

Minor Weaknesses / Questions:

 - I don't really consider your "Problem Statement" to be a problem statement. It is more of a description of the method and system level architecture.

 - Further, I find the definition of Problem 1 odd. The optimization is solvable for any choice of $D_{new}$. Shouldn't the optimization be formulated to be over $D_new$ explicitly?

 - Why did you choose the specific data augmentations that you did as a baseline. They seem like reasonable choices for the airline taxi case, but not necessarily for the other two.

**Summary Of Recommendation:**

Although I believe the work to have high potential, I have significant concerns about the method as outlined above. If the authors are able to justify their choices then I am open to improving the score but at present they seem like fundamental flaws (particularly the choice of label in the adversarial data).

---

### Meta-Review · Area_Chair_WqMo · 2022-08-09

**Recommendation:** Accept (Poster)
**Confidence:** 4

**Metareview:**

This paper proposes to generate task-specific data augmentation to improve the robustness of vision-based robotic control. The adversarial dataset for augmentation is synthesized based on the sensitivity analysis of a differential MPC framework. The paper shows that these adversarial datasets improve the performance of the controllers on OoD test scenarios in three simulated domains: navigation, manipulation and drones.

Most of the reviewers agree that the proposed idea is interesting; the end-to-end differentiability of MPC enables easy adversarial scene generation; and the evaluations are reasonably thorough. The reviewers also brought up a few important areas for improvement, including validity of assumptions (Reviewer 2DMp), more baselines (Reviewer bY11), evaluations in complex scenes such as the real world (Reviewer rbua), trade-off between small performance gain vs. the increased complexity (Reviewer rbua), and quality of writing for the Limitation Section (Reviewer bmgX).

This paper received a lot of discussion during the reviewer discussion phase. The focus of debate was the strong assumption that adversarial labels are equal to the unperturbed input, which may significantly limits the application domain. Reviewers agreed that this approach would still be useful for some tasks, such as the X-plane example where the same label as the training data can be used. More importantly, the paper was written in a way that this important limitation was buries into details. In the future version of this paper, please revise it to be more explicit about the assumptions and the limitations. Reviewer generally agreed that the benefits of publishing this paper would outweigh the limitations, thanks to the interesting idea and thorough evaluations. For this reason, we recommend accepting the paper.

**Best Paper Nomination:**

No